# A Comprehensive Review of Hardware Acceleration Techniques and Convolutional Neural Networks for EEG Signals

**DOI:** 10.3390/s24175813

**Published:** 2024-09-07

**Authors:** Yu Xie, Stefan Oniga

**Affiliations:** 1Faculty of Informatics, University of Debrecen, 4032 Debrecen, Hungary; yu.xie@inf.unideb.hu; 2North University Center of Baia Mare, Technical University of Cluj-Napoca, 400114 Cluj-Napoca, Romania

**Keywords:** EEG signal analysis, convolutional neural networks, feature extraction, hardware acceleration

## Abstract

This paper comprehensively reviews hardware acceleration techniques and the deployment of convolutional neural networks (CNNs) for analyzing electroencephalogram (EEG) signals across various application areas, including emotion classification, motor imagery, epilepsy detection, and sleep monitoring. Previous reviews on EEG have mainly focused on software solutions. However, these reviews often overlook key challenges associated with hardware implementation, such as scenarios that require a small size, low power, high security, and high accuracy. This paper discusses the challenges and opportunities of hardware acceleration for wearable EEG devices by focusing on these aspects. Specifically, this review classifies EEG signal features into five groups and discusses hardware implementation solutions for each category in detail, providing insights into the most suitable hardware acceleration strategies for various application scenarios. In addition, it explores the complexity of efficient CNN architectures for EEG signals, including techniques such as pruning, quantization, tensor decomposition, knowledge distillation, and neural architecture search. To the best of our knowledge, this is the first systematic review that combines CNN hardware solutions with EEG signal processing. By providing a comprehensive analysis of current challenges and a roadmap for future research, this paper provides a new perspective on the ongoing development of hardware-accelerated EEG systems.

## 1. Introduction

Electroencephalography (EEG) has become a fundamental area of neuroscience research, clinical diagnostics, and brain–computer interface (BCI) applications. EEG signals can record the activity of the brain’s electrical waves and are a key area of interest for practitioners studying neural activity in the human brain and the BCI. Applications of EEG include emotion classification, motor imagery, epilepsy detection, sleep monitoring, attention monitoring, cognitive load assessment, BCI control systems, neurofeedback training, detection of pathological brain activity, and diagnosis of mental disorders. These applications include but are not limited to medical diagnosis, neurorehabilitation, mental health monitoring, education and learning assessment, augmented and virtual reality interaction, user status monitoring, and many other fields. As the relevance of EEG extends across various domains, there is a growing demand for proficient and efficient signal-processing techniques to fully harness the latent capabilities of EEG data. Previous reviews related to EEG have mostly focused on software development. For example, these reviews covered general deep learning methods based on EEG [1], analyzed transfer learning mechanisms for common EEG tasks [1], and listed existing software solutions [2]. However, beyond these software-focused discussions, it is also crucial to consider application scenarios that demand small size, low power consumption, high security, and high accuracy, such as motor imagery and emotion detection. The authors of [3] briefly discussed hardware solutions for EEG-based emotion recognition and introduced classifiers like support vector machines (SVMs), k-nearest neighbor (KNN), and artificial neural networks (ANNs), providing a performance comparison. Unfortunately, they did not extensively cover the hardware design of deep learning methods. Despite some similarities, our work has the following distinct features:(1)This paper focuses on the challenges of hardware acceleration in EEG-based systems, which we believe is a critical area in the development of wearable medical solutions.(2)We conduct a systematic analysis of EEG signal characteristics and categorize existing feature extraction methods, which helps identify the most suitable hardware acceleration strategies for different application scenarios and provides a clear direction for further research.(3)To the best of our knowledge, this paper is the first to systematically review the combination of CNN hardware solutions with EEG signal processing, filling a gap in this specialized field.(4)This paper comprehensively considers the development of EEG and CNN technologies, identifies existing challenges, and provides insights into future development directions, offering a new perspective for research in this area.

### 1.1. EEG Signal-Processing Challenges

The characteristics of EEG signals include high temporal resolution and a direct reflection of neural activity [4]. However, they present unique challenges due to their low spatial resolution, sensitivity to noise, and high dimensionality. Extracting meaningful information from EEG data necessitates complex signal-processing techniques. EEG signal processing generally includes preprocessing, parameter determination, and feature extraction. There is currently no established set of paradigms for EEG signal processing.

Non-deep learning methods commonly used in EEG signal processing include techniques such as SVM, kNN, and Linear Discriminant Analysis (LDA), as well as traditional feature extraction approaches like Independent Component Analysis (ICA) and Principal Component Analysis (PCA). While these non-deep learning methods are indeed simpler, faster, and often easier to implement on wearable devices, they may struggle to achieve high accuracy due to the inherent complexity and continuous nature of EEG signals. These traditional methods typically rely on handcrafted features, the selection of which is highly dependent on the researcher’s expertise and can be time-consuming. Additionally, these models often have limited analytical power, which may not be sufficient for handling complex tasks. In contrast, deep learning greatly simplifies the processing workflow by automatically extracting features while simultaneously performing decoding. Although deep learning models do involve a large number of parameters, resulting in significant computational demands, this is precisely where hardware acceleration becomes particularly valuable. Hardware acceleration enables the efficient execution of these models, making it feasible to deploy deep learning approaches in real-time EEG signal-processing applications, even in resource-constrained environments. Recent advances in deep learning have demonstrated remarkable capabilities in various fields, such as image processing and natural language understanding [5]. These advancements have spurred interest in utilizing deep learning for EEG signal analysis. Convolutional neural networks (CNNs) are one of the most common applications of deep learning, particularly in image processing. Images, as a form of feature representation, allow for the visualization of EEG signals’ temporal and frequency domain characteristics through additional spatial information. This enables CNNs to fully exploit the characteristics of EEG data contained in the input. In addition, CNNs can autonomously learn discriminative features from EEG data, potentially reducing reliance on manual feature engineering and enhancing classification performance. It is, therefore, natural for EEG researchers to attempt to replicate this approach.

### 1.2. Hardware Acceleration and Efficiency

While CNNs offer promising solutions for EEG signal analysis, they come with substantial computational demands, particularly in real-time applications like BCIs. One possible approach is to deploy processing tasks to cloud servers with more powerful resources [6,7]. But in many cases, applications such as real-time emotion monitoring and motor imagery need to respond immediately to changes in the user’s state. Relying on cloud servers introduces network latency. If there is network fluctuation or no network, the real-time requirements may not be met. In addition, EEG data usually contain sensitive information, and transmitting these data to the cloud for processing may raise privacy and security concerns. Therefore, for many wearable EEG devices, hardware acceleration is necessary for achieving key functions such as low power consumption, real-time processing, data security, and independent operation. These factors make hardware acceleration particularly important for wearable devices. Hardware acceleration, especially on Field-Programmable Gate Arrays (FPGAs) and Graphics Processing Units (GPUs), can significantly enhance the speed and efficiency of EEG processing based on CNNs [8]. FPGAs, in particular, provide a reconfigurable hardware platform that allows researchers to optimize CNN architectures for specific EEG applications. FPGA-based implementations are renowned for their parallel processing capabilities, low latency, and energy efficiency. These characteristics make FPGAs an excellent choice for real-time EEG processing [9].

### 1.3. Research Objectives and Structure

The primary objective of this paper is to provide a comprehensive review of the integration of CNNs and hardware acceleration techniques for EEG signal processing. To achieve this goal, we have structured this paper into several sections. Section 2 reviews EEG-based hardware solutions, including preprocessing, feature extraction, and various features used in EEG analysis. Section 3 discusses efficient CNN architectures, including pruning, quantization, tensor decomposition, knowledge distillation, and neural architecture search. Section 4 explores the challenges in this field and provides recommendations for future research directions. Section 5 summarizes the essential findings and insights of this paper. The purpose of this review is to explore the latest achievements in the field of EEG and CNNs and to help more researchers gain a deeper understanding of the hardware solutions, potentials, and challenges in this field.

## 2. EEG-Based Hardware Solutions

EEG signals are variable and contain a lot of noise. As researchers study EEG, they have developed many algorithms to extract raw signals and convert them into usable information. This process comprises several critical stages, each employing a unique set of techniques and technologies. In this section, we present the state-of-the-art hardware implementations of various algorithms used in the preprocessing and feature extraction stages.

### 2.1. Preprocessing

Before exploring the wealth of information hidden within EEG signals, it is typically necessary to preprocess the data to address the substantial noise and artifacts that often accompany them [10,11]. While some datasets offer EEG data with certain artifacts already removed, the hardware implementation of preprocessing blocks provides a more comprehensive and practical evaluation of the hardware resources required for EEG-based solutions.

These artifacts can be both physiological and non-physiological. Physiological artifacts are associated with activities such as head movements (EMG), blinking (EOG), muscle contractions, heart activity (ECG), sweating, and respiration (RS) [12,13]. Non-physiological artifacts occur due to external factors like electrode malfunctions (Ep), power sources (AC), ventilation, magnetic fields, cable movements (Cable), and more [14,15]. As summarized in Table 1, the spectrum of EEG signals spans from 0 to 100 Hz and intertwines with various artifacts across this range.

The frequency range of EEG bands is defined slightly differently in different studies. It is worth noting that in Table 1, the frequency bands of Mu waves and Alpha waves overlap because the Mu band is a special form of the Alpha band. Alpha waves are among the most easily recognizable in EEG and are usually found in the posterior region of the head (occipital lobe) when the subject is relaxed or has their eyes closed. Mu waves usually appear in the sensory-motor area of the human brain. When the subject is at rest, Mu waves can be clearly observed; however, when the subject is moving or exercising, the amplitude of the Mu wave decreases or disappears, which is called motor inhibition. Although the Alpha and Mu bands are very similar, they come from different brain regions and have distinct associated functions. The authors of [16] showed that ICA can help distinguish activities from different brain regions, thereby differentiating Alpha waves from Mu waves.

One method for reducing noise in EEG signals is the application of digital filters. In [17], a bandpass filter ranging from 8 to 45 Hz was used. While this configuration successfully eliminated some low-frequency artifacts, it came at the cost of attenuating EEG activity in the Theta and Delta brainwaves. In contrast, [18,19] employed a low cutoff frequency of 2 Hz. This lenient setting allowed for the coexistence of slow eye-movement artifacts with the EEG recordings. Ref. [20] used an 81-order digital filter, which provided a better graphical representation for the analysis of Beta and Mu brainwaves and was successfully implemented on the DE0-Nano-Soc development card. It is worth noting that using only bandpass filtering may not cover complex interference scenarios caused by actual artifacts. Another method was described in [21], which involves three key steps: downsampling, bandpass filtering, and integrating a Mean Average Reference (MAR) filtering block. Downsampling, performed at twice the rate without altering the frequency content, reduces the power spectral amplitude by half and enhances signal clarity. The bandpass filter operates within the range of 2 Hz to 60 Hz, ensuring that EEG signals remain within the desired spectrum. The MAR filter involves re-referencing electrode signals relative to their collective average, effectively reducing the signal’s dependence on specific electrode locations [4]. This is achieved by subtracting the average signal value from each electrode signal. With this technique, artifacts associated with the physical reference electrode are effectively removed, while averaging acts as noise suppression, preventing the introduction of new artifacts into the EEG signal.

On the other hand, in [22], the authors initiated an adaptation strategy employing ICA based on kurtosis. This technique can extract artifacts co-occurring within the same frequency band as EEG signals. However, the application of this method is entirely offline, and the feasibility of its hardware implementation remains unconfirmed. In [23], fast ICA was employed for epileptic seizure detection from multi-channel signals. This rapid ICA assists in artifact removal and reduces energy dissipation. Eigenvalue decomposition (EVD), used in conjunction with the Jacobi algorithm, resulted in a 77.2% reduction in area. The practical performance of this architecture was implemented on an FPGA and validated through appropriate databases. These blind source separation algorithms have become popular methods for hardware implementations of EEG preprocessing [24,25].

Nevertheless, the efficacy of these methods is related to the number of independent components obtained, a quantity dependent on the number of signal sources or electrodes. Consequently, these methods exhibit limitations in situations with a small number of sources. This realization has prompted the exploration of cost-effective alternatives [26,27] for integrating the ICA model, where the authors developed a performance evaluation method that fused matched filtering and threshold cross-checking based on the number of electrodes.

In the field of EEG-based classification, effective methods must encompass a set of techniques to ensure signal quality, including but not limited to detrending, re-referencing, outlier detection, interpolation, detection and correction, and elimination of artifacts.

### 2.2. Feature Extraction

Preprocessed EEG data include a wealth of information. However, to use this information, it must first be extracted and converted into meaningful features. This section reviews feature extraction techniques that transform EEG signals into quantifiable attributes. We examine five types of feature extraction used in this domain: frequency-based features, time-based features, time-frequency features, spatial features, and raw signals.

#### 2.2.1. Frequency-Based Features

Frequency-based features analyze the rhythmic patterns of brain activity. Given that a large number of studies have identified correlations between spectral components and brainwave rhythms, researchers have a clear tendency to focus on frequency characteristics [5,28].

The complexity of related algorithms varies greatly across software platforms, and researchers rarely consider the issue of computing resource consumption at the software level. Consequently, their high complexity levels make many of them unsuitable for direct hardware implementation. However, methods such as Power Spectral Density (PSD), Fourier Transform, Discrete Cosine Transform (DCT), Singular Spectrum Analysis (SSA), and combinations of filter banks still hold promise for integration into hardware solutions, as the hardware design of these traditional signal-processing modules is well established. Hardware-accelerated Fast Fourier Transform (FFT) modules, with their moderate resource requirements, can efficiently compute PSD. In contrast, higher-order spectral (HOS) analysis, Linear-Frequency Cepstral Coefficients (LFCCs), Mel-Frequency Cepstral Coefficients (MFCCs), the Burg method, and Auto-Regressive Reflection Coefficients (ARRCs), while effective, might require more resource allocation and design work for successful integration into hardware solutions.

Cross-frequency coupling (CFC) functions are promising approaches that leverage synchronized variations across different frequency bands [29,30]. These functions manifest as connectivity graphs or indices. They not only ensure robustness but also have the potential to reduce EEG data rates and memory requirements. However, the computational complexity of CFC algorithms and the hardware demands, such as Coordinate Rotation Digital Computer (CORDIC) processors for computing cross-frequency Hilbert transforms, pose significant challenges for ultra-low-power hardware designs [31].

It is foreseeable that PSD has become one of the most popular methods in the EEG classification domain. In [18,21], PSD approximations were used, with only one power value per channel calculated based on the average activity within the beta band (12 to 30 Hz). While this approach does not necessitate FFT accelerators, making it more hardware-friendly, it introduces information loss within the frequency bands. In contrast, [22] employed an FFT with 128 frequency bins, ensuring sufficient resolution to capture fluctuations both within and between the EEG wavebands.

#### 2.2.2. Time-Based Features

In addition to frequency-based features, time-based features are also a key research direction in EEG signal processing because they can reflect the dynamic changes in brain activity over time and provide indicators, such as maximum and average amplitudes, slopes, and skewness, that reflect transient brain activity.

Notably, Higher-Order Crossings (HOCs) have become a common approach. HOCs are based on tracking zero crossings and their consecutive differences, offering valuable insights into the temporal dynamics of EEG signals. Higuchi’s Fractal Dimension (HFD) is well known for its ability to capture the non-linear aspects of data, making it particularly suitable for feature extraction in various EEG events. Other time-based feature extraction methods include Hjorth parameters, Grassberger–Procaccia (GP), Hybrid Adaptive Filtering (HAF), the Hurst exponent, Local Binary Patterns (LBPs), and Event-Related Late Positive Potentials (LPPs).

Compared to frequency-domain approaches, the advantage of time-based methods is their relatively lower hardware resource requirements, making them an attractive choice for hardware implementations. However, it must be acknowledged that methods involving non-linear transformations, such as HFD or Hjorth parameters, often increase the complexity of hardware design. In [19], a time-based approach was employed for feature extraction. The method focuses on extracting HOCs and skewness (SK) information from time series to construct feature vectors. To facilitate hardware-friendly implementation, [19] introduced the Approximate SK Indicator (ASKI), significantly reducing gate counts by a factor of 86. For HOCs, the authors employed a straightforward implementation based on comparators, using 3-bit SR triggers to monitor the previous values of a given sample. Identifying sign changes between consecutive sample values triggers a simple counter to accumulate zero crossings. Both of these metrics are standardized and fed into a dense neural network. Hjorth parameters indicate statistical characteristics in EEG signal analysis. Calculating them requires variance computation for the “activity” parameter, the square root of the ratio for the “mobility” parameter, and division operations for the “complexity” parameter with first-order derivatives [32].

Sample Entropy (SE) is another time-based method [33] that has proven to be relatively hardware-friendly due to its simplicity. SE is a measure of time-series complexity defined as the logarithm of a ratio, making its hardware implementation relatively straightforward. Ref. [34] proposed that SE architecture runs ten times faster on FPGAs than on CPUs. Methods with minimal hardware resource utilization and power consumption include statistical mean and other time-based methods like Root Mean Square (RMS), Variance (Var), and Standard Deviation (SD), as observed in the articles we surveyed, both on software and hardware platforms, these methods all have an average accuracy of no more than 60%. Therefore, they are the least ideal options.

#### 2.2.3. Time-Frequency Features

The fusion of time-domain and frequency-domain techniques gives rise to time-frequency features, which are adept at handling the non-stationary nature of EEG data. Wavelet Transforms (WTs) and their various forms (such as Discrete Wavelet Transform (DWT), Continuous Wavelet Transform (CWT), and Adaptive Tunable Q-Wavelet Transform) play a significant role. The Short-Time Fourier Transform (STFT) and the Hilbert–Huang Transform (HHT), along with its Independent Mode Function (IMF) and Empirical Mode Decomposition (EMD), have also garnered attention. Unlike Fourier and Wavelet Transforms, the HHT employs a unique approach using EMD to decompose signals, extracting quasi-harmonics starting from the highest frequency. This method demonstrates robustness in analyzing non-stationary and non-linear time series, a common characteristic of EEG signals [35]. However, it is worth noting that the calculation of the IMF in the HHT involves an iterative algorithm. The parameters of each layer of this algorithm are highly correlated, making it difficult to take advantage of hardware parallel processing, especially in some real-time processing application scenarios.

Wavelet Transforms (WTs) and fundamental forms of the STFT are two feasible choices for hardware implementation. Ref. [36] proposed an STFT design based on the Xilinx Zedboard and extracted 20 frequency feature components for classification. These 20 features were divided into five groups, each corresponding to five different EEG wave frequency bands. Features were extracted from 128 channels of EEG signals at a clock rate of 50 MHz. Ref. [37] employed the STFT for time-frequency analysis of EEG segments and extracted the regions of interest in terms of frequency bands and features, implementing a deep learning model and an STFT module on an FPGA to enhance seizure detection. Ref. [38] introduced a generalized spectrogram method. The STFT was applied to signals captured from six electrodes, which were subsequently mixed based on asymmetric indices. Ref. [39] implemented a DWT with a two-class LDA classification on the Basys 3 Artix-7 FPGA Board. Ref. [40] presented a design and implementation of the Morlet CWT for EEG signals on the Spartan 3AN FPGA. By optimizing the trade-off between speed and silicon area, it could generate all-scale wavelet coefficients for a 1024-point EEG signal in approximately one millisecond when designed to run at a maximum clock speed of 125 MHz. In the STFT, the signal to be processed is divided into multiple shorter signals according to the size of the window, and a separate FFT is then applied to each small signal. In contrast, the WT processes the complete signal. Both methods yield similar results, with the Wavelet Transform being particularly effective in handling non-periodic and fast-changing features. In contrast, the STFT is favored for real-time processing due to its shorter processing time windows. Moreover, advances in FFT hardware acceleration and various space-efficient techniques for compressed FFT accelerators make them highly attractive for hardware-based solutions.

#### 2.2.4. Spatial Features

Unlike independent features, spatial features depend on the availability of time-based, frequency-based, or time-frequency EEG data for computation. This section introduces a well-known spatial feature method: Common Spatial Pattern (CSP). In a two-classification problem, CSP aims to find spatial filters that maximize the variance of EEG signals for one class while minimizing it for the other class. This results in spatial filters that enhance discriminative features for the two classes, making classification more effective. Here, a simplified explanation of the formula application of CSP is presented. The spatial filter w is calculated by simultaneous diagonalization of the sample covariance matrices from both classes, as follows:(1)$$J\left(w\right)=\frac{w^{T}\bar{C_{1}}w}{w^{T}\bar{C_{2}}w}$$
where *T* represents the transpose. $\bar{C_{1}}$ and $\bar{C_{2}}$ denote the average covariance matrices of the two different conditions or classes, respectively, and are defined as follows:(2)$$\bar{C_{k}}=\frac{1}{N}\sum_{n=1}^{N_{k}}\frac{D_{\left(k,n\right)}D_{\left(k,n\right)}^{T}}{Trace\left(D_{\left(k,n\right)}D_{\left(k,n\right)}^{T}\right)},\;k=1,\;2$$
where $Trace\left(D_{\left(k,n\right)}D_{\left(k,n\right)}^{T}\right)$ represents the solution of the matrix trace. $N_{k}$ is the number of samples for the *k*th class, that is, the number of single-trial data. $D_{\left(k,n\right)}$ represents the *n*th trial data of the *k*th class. The CSP algorithm frames this as a generalized eigenvalue problem: (3)$${\bar{C_{2}}}^{-1}\bar{C_{1}}w=\;\lambda w\;$$

The conventional CSP feature extraction method derives features by computing the logarithm of variances from spatially filtered signals, as follows:(4)$$f_{p}=\;\log\left\{\frac{var\left(Z_{p}\right)}{\sum_{i=1}^{2m}var\left(Z_{i}\right)}\right\}\;,\;p=1,\;2,\;\cdots ,\;2m\;$$
where the first and last *m* rows of the EEG signals are usually taken as the final spatial filter. Here, *Z* is given by $Z=w^{T}D$.

In [41], the CSP model was implemented on an Altera Stratix-IV FPGA and verified on the BCI competition dataset. In [42], CSP was used as the feature extraction algorithm with Mahalanobis Distance (MD) as the classifier, and the entire signal-processing task was executed on software embedded in a Nios II processor within a Stratix IV FPGA. This approach reduced FPGA resource consumption but introduced a time delay of 0.5 s. An improvement on the approach in [41] can be found in [42], where the authors accelerated some functions in hardware but still performed most processing on the Nios II, reducing the delay to 0.399 s. In [43], the authors proposed a complete hardware implementation of a BCI embedded on an FPGA using Scalp-Coupled Common Spatial Spectral Patterns (SCSSPs) as the feature extraction algorithm and support vector machines (SVMs) as the classifier.

The previous section mentioned the use of asymmetric indices as a subset of spatial features. These indices are used to compare the relationship between the two hemispheres and thus indicate asymmetric neural activity [44]. This method depends on the calculation of asymmetric indices, which are usually computed between electrode pairs and can be effectively used alongside frequency, time, or time-frequency features. Hardware designs, such as those in [18,21], combine asymmetric indices with channel-based PSD averages, a concept discussed earlier in the frequency-based feature category. In these implementations, utilizing eight channels, the asymmetric indices are calculated based on the PSD averages of electrode pairs.

In addition to CSP and asymmetric indices, another subset of spatial-based features (specific to EEG-based emotion detection systems implemented in software) includes connectivity features, particularly Directed Transfer Function (DTF) features. DTF features serve as causal measurements used to determine brain connectivity patterns [45]. The DTF quantifies the causal influence of one EEG channel on another at specific frequencies. While theoretically, the DTF can be implemented in hardware, it presents more complex challenges compared to asymmetric indices. This complexity arises from the need to calculate the transfer matrices of multivariate autoregressive models and to perform square root and division operations.

#### 2.2.5. Raw Signals

This section introduces some works on EEG classification using raw signals. Due to the complexity of EEG signals, it is difficult for traditional EEG classifiers to obtain good results from raw signals. The shift in researchers’ views is mainly due to the popularity of deep neural network (DNN) classifiers. In these new DNN approaches, raw EEG signals can be directly input into the network for feature extraction and classification. Currently, there are no hardware-based implementations that follow this paradigm. This is mainly due to the need to compensate for the lack of specific features in the complexity of DNNs [46,47]. Moreover, most DNN hardware implementations for EEG tend to adopt shallow DNNs to achieve designs that save more computing resources and space. These networks lack the direct capability to process raw EEG data. However, some new material technologies may change this status, such as near/super-threshold techniques in Fully Depleted Silicon on Insulator (FDSOI) technology [48] or adaptive body biasing in advanced semiconductor technologies [49]. These technologies have the potential to facilitate hardware implementations of DNNs that can directly operate on raw EEG signals in wearable classifiers.

Another approach is to use a DNN only as a feature extractor instead of as a classifier. An example can be found in [50], where overlapping filters, as well as Differential Entropy (DE) and PSD analysis, were employed to derive suitable kernels that guide a CNN in automatically extracting features from raw EEG data. The advantage of this approach lies in implicitly merging spectral and entropy information when dealing with raw EEG data for learning and classification.

Table 2 shows the methods discussed in this section.

## 3. Efficient Convolutional Neural Networks

In Section 2, we mentioned that DNNs have been widely used in the study of EEG signals. CNNs have showcased remarkable capabilities in various domains, from image recognition to natural language understanding. Their innate ability to autonomously learn hierarchical features has positioned them as the predominant force in software-driven DNNs. The application of CNNs presents an opportunity to fundamentally reshape the approach to neural data analysis, interpretation, and classification. It is well recognized that the surprising progress of CNNs has been accompanied by a large number of parameters that need to be calculated. In particular, the real-time calculation of CNNs is challenging in the field of BCIs. This section introduces five highly efficient CNN hardware techniques specifically used for EEG signal processing: pruning, quantization, tensor decomposition, knowledge distillation, and neural architecture search. The applications we reviewed are summarized at the end of this section.

### 3.1. Pruning

One approach to reducing the computational complexity of CNNs is pruning, a technique dating back to the early 1990s [51]. Researchers have found that there are some redundant or less informative neural nodes in CNNs. Therefore, deleting these unimportant weights, filters, channels, and even layers during the inference process has little effect on accuracy. The core idea of pruning is to identify and eliminate redundancy in the network to reduce computational and memory requirements. This process can be categorized into two main strategies: weight pruning and structural pruning.

#### 3.1.1. Weight Pruning

Weight pruning is a core technique in the design of efficient neural networks. Removing some unimportant connections does not affect the performance of the network because not all neural nodes are equally important. Several strategies have been developed to effectively execute weight pruning.

Early work in this field, such as Optimal Brain Damage (OBD) [51] and Optimal Brain Surgeon (OBS) [52], introduced the concept of utilizing second-order derivatives (Hessian matrix) of the loss function to identify unnecessary weights. These methods laid the foundation for subsequent weight-pruning techniques.

A widely adopted three-step deep learning method was proposed in [53,54]. First, the network is initially trained to identify essential connections. Subsequently, less important connections are pruned. Finally, the network undergoes retraining to fine-tune the remaining connections. This iterative process gradually leads to a more compressed network structure. Furthermore, dead neurons, with all their connected weight pruned, can be safely discarded. Some research has suggested that pruned larger models can outperform smaller dense models in terms of accuracy under the same memory constraints [55,56]. While traditional weight pruning methods operate in the spatial domain, [57] proposed a frequency-domain approach that transforms spatial weight into frequency-domain coefficients. Pruning is performed dynamically in different frequency bands during each iteration, leading to higher compression ratios.

In the previously mentioned methods, a global magnitude threshold is applied to the entire DNN, which can affect network performance since applying the same compression rate to each layer is not always optimal. Some studies have used variable pruning rates to target layers of different importance. Ref. [51] combined selective learning, identifying weight importance concerning the loss, and discarding other weights by cutting off the gradient flow. Ref. [58] employed a Gaussian Mixture Model (GMM) for each layer’s weight distribution to determine layer-wise compression rates. Pruning layers are selected based on the number of weights, with small amplitudes estimated using the GMM.

Additionally, amplitude-aware pruning methods like [59] introduce considerations for energy reduction along with compression ratios and accuracy loss when determining pruning strategies.

Weight-pruning techniques remove less important weights or neurons, regardless of their position within the network. This process results in so-called “unstructured” networks. Such networks may require specialized software or hardware adaptations, thus limiting the universality of the weight-pruning algorithm on hardware.

#### 3.1.2. Structural Pruning

On the other hand, structural pruning focuses on eliminating entire filters, channels, or even whole layers. This approach not only reduces computational requirements but also leads to more compact network architectures, which is especially valuable in resource-constrained environments. Unlike weight pruning, structural pruning does not require specialized hardware or software support. Because it only changes the size of the input and output, it does not change the parameters within the layer. This method maintains strong compatibility. Mainstream structural pruning strategies primarily aim to identify and remove unimportant filters. They are typically categorized into three main branches:Criterion-Based Pruning: In this method, filters are ranked based on specific pruning criteria. Methods that consider both L1 and L2 norm criteria, such as L1-norm-based pruning [60] and Soft Filter Pruning (SFP) [61], have been explored. Additionally, activation neuron criteria, such as the Average Percentage of Zeros (APoZ) [62], help identify and remove filters with low APoZ values.Ranking-Based Pruning: HRank [63] employs feature map ranking as the metric for filter pruning. Lower-ranked feature maps are considered to contain less information, making them one of the primary candidates for pruning.Optimization-Based Pruning: Methods like ThiNet [64] and NISP [65] formulate pruning as an optimization problem. ThiNet prunes filters in the current layer while minimizing the reconstruction error in the next layer. On the other hand, NISP focuses on reducing the reconstruction error in the “Final Response Layer” (FRL).

In addition to filter and channel pruning, structural pruning extends to whole-layer pruning methods [66,67,68]. These methods selectively remove layers from the network using various criteria, resulting in more compact models. Similarly, these pruned models not only reduce the amount of computation and time but also maintain good accuracy.

The above algorithms all follow the conventional format of training models, pruning, and fine-tuning, except for the method in [69]. In structural pruning, fine-tuning pruned models may produce performance comparable to or even worse than training the same model from scratch with randomly initialized weights. Innovative techniques like network slimming [70] and group LASSO loss [71] automatically identify and prune unimportant channels or neurons during the training process, effectively achieving model compression without the need for post-pruning fine-tuning.

### 3.2. Quantization

Quantization is another technique aimed at improving the efficiency of CNNs. It reduces the precision of network weights and activations, typically from high-precision formats (e.g., FP32) to lower-bit fixed-point representations. Precision reduction helps lower memory requirements and computational complexity, making CNNs more hardware-friendly. A commonly used CNN hardware implementation strategy for EEG signal processing is to use quantization and pruning together to achieve better computational efficiency.

In the quantization process, neural network data are constrained to a set of discrete levels, offering various distribution options, including uniform or non-uniform distributions. Uniform distribution [72] (characterized by even step sizes) and non-uniform distribution [73] (typically represented by logarithmic distributions) are commonly used quantization schemes. Quantization can be achieved using deterministic methods and stochastic methods. Deterministic methods (such as projecting data onto the nearest discrete levels [71]) ensure precision, while stochastic methods involve probabilistically determining which of the two neighboring discrete levels the data are projected onto.

The roots of quantizing neural network data can be traced back to the early 1990s [74]. This technology offers two primary forms: post-training quantization [75,76] and quantization-aware training [77,78,79]. In post-training quantization, FP32 weights and activations are converted to lower-precision formats after the model’s full training. Conversely, quantization-aware training incorporates the quantization error as part of the training loss, typically resulting in improved model accuracy.

The field of quantization has achieved impressive results. Research has indicated that FP32 precision parameters can be effectively reduced to INT8 without significant loss of accuracy [75]. Some researchers have even developed 4-bit quantization approaches, eliminating the need for fine-tuning post-quantization [76]. Additionally, experiments using INT8 precision for training and inference have shown competitive results, such as a 1.5% accuracy loss in the case of the ResNet-50 model [77]. Furthermore, advancements like generalized bit precision [78] provide greater flexibility by allowing different bit depths for storing weights and activations, offering more flexibility than traditional INT8 quantization. For example, [78] used 1-bit weights and 2-bit activations to quantize AlexNet, achieving a top-1 accuracy of 51%.

Quantization-aware INT8 training methods [79] enhance the practicality and performance of quantization in deep neural networks by cleverly optimizing calculations in both forward and backward passes through the thoughtful incorporation of loss-aware compensation and parameterized range clipping.

Within the scope of quantization strategies, binarization represents the most extreme form, reducing data to only two possible values: (0, 1) or (−1, 1). This radical reduction allows for resource-efficient XNOR and bit-count operations instead of resource-intensive matrix multiplications. Therefore, binarization is particularly suitable for deploying deep neural networks on resource-constrained devices. However, this extreme quantization inevitably results in significant information loss. Additionally, the discontinuous nature of binary values presents optimization challenges for binary neural networks.

In recent years, to address these challenges, researchers have developed several promising algorithms. Innovative methods like BinaryConnect [80], binary neural networks (BNNs) [81], and XNOR-Net [82] have garnered attention in the field of binary neural networks. BinaryConnect introduces random techniques for binarizing neural networks, applying binarization during both forward and backward propagation. BNNs extend BinaryConnect by binarizing activations, marking a significant milestone in binary neural network research. Both BinaryConnect and BNNs employ combinations of deterministic and stochastic binarization functions to simplify hardware implementations. In contrast, XNOR-Net employs a different approach, using scaling factors for binary parameters to approximate floating-point parameters. This allows the weight quantization in XNOR-Net to be represented as w≈α∗bw, where α represents the floating-point scaling factor of the binary weight bw.

Recent advancements include optimization-based binarization techniques. Methods like XNOR-Net and DoReFa-Net [72] concentrate on mitigating quantization errors during the training process. For instance, DoReFa-Net quantizes gradients to expedite training. In addition to local layer considerations, other binarization techniques, such as loss-aware binarization [83] and incremental network quantization [84], directly minimize the overall binary weight loss across the entire network. Through the application of an adaptive error decay estimator, IR-Net [85] represents a pioneering solution for preserving information flow during forward and backward propagation.

Different quantization strategies provide researchers with more options, requiring them to strike a balance between computational efficiency and model accuracy based on specific tasks in EEG signal processing.

### 3.3. Tensor Decomposition

Tensor decomposition is a technique aimed at breaking down weight tensors into smaller tensors or matrices. This process significantly reduces the number of parameters in CNNs while preserving accuracy. DNN parameters within convolutional layer weight tensors and fully connected layer weight matrices typically exhibit low-rank characteristics [86]. Based on this observation, the directions of algorithm design by researchers can be categorized into two main classes: low-rank matrix decomposition and tensor decomposition.

#### 3.3.1. Low-Rank Matrix Decomposition

The core of low-rank matrix decomposition techniques lies in approximating the weight matrices of DNN layers by multiplying several low-rank matrices. Singular Value Decomposition (SVD) [87] is the most popular low-rank approximation method, emphasizing the description of a matrix through its singular values.

SVD decomposes a matrix A∈R into three key components:(5)A=UDV
where U∈R and V∈R are orthogonal matrices, and D∈R is a diagonal matrix containing singular values. A compact and efficient network model is achieved by retaining only the most critical components in the decomposition matrices.

In practice, SVD-based techniques have played a crucial role in various applications. For instance, in [88], SVD was utilized to decompose the product of weight matrices and input data, further enhancing model efficiency. The authors of [89] introduced sparsity into low-rank factorized matrices by maintaining a lower rank for less critical neurons, thus improving compression rates. Additionally, in [90], channel-wise SVD decomposition of convolutional layers was employed to enhance efficiency by subdividing kernels into two consecutive layers of different sizes.

Using SVD at every training node results in high computational costs. Therefore, it becomes important to determine the importance of each DNN layer during training. Innovative approaches have emerged to address these issues. In [91], an SVD training technique was proposed to explicitly achieve low-rank approximation without invoking SVD at every training step. Furthermore, [92] introduced a joint matrix factorization scheme that simultaneously decomposes layers with similar structures and optimizes based on SVD.

#### 3.3.2. Tensorized Decomposition

Tensors, multi-way data arrays, provide a universal foundation for representing complex data. In the context of deep learning, these tensors encompass various orders, with two-dimensional matrices representing second-order tensors commonly used in fully connected layers. Meanwhile, the weight tensors of convolutional layers are represented as fourth-order tensors. Higher-rank tensors require higher compression ratios. Tensor decomposition extends the idea of low-rank matrix decomposition to high-order tensors, breaking down weight tensors into smaller tensors and thus reducing the network’s complexity. Prominent methods in this field include Tucker decomposition [93], CANDECOMP/PARAFAC (CP) decomposition [94,95], tensor train (TT) decomposition [96,97,98], and tensor ring (TR) decomposition [99]. In [93], Tucker decomposition was used to compress convolutional weight kernels with predefined ranks. Simultaneously, the CP decomposition strategically breaks down convolutional kernels into a collection of first-order tensors, reducing parameters and training time. This method requires an iterative process that cleverly combines decomposition with fine-tuning for each convolutional layer. Refs. [94,95] differ in whether they decompose all convolutional layers. The former only decomposes a few convolutional layers, while the latter decomposes all convolutional layers. Ref. [100] introduced a robust low-rank decomposition technique that merges Tucker decomposition with CP decomposition. In this approach, the core tensors extracted through Tucker decomposition are further decomposed using CP decomposition. Other tensor decomposition techniques, such as TT decomposition, appear to be more favorable for recurrent neural networks (RNNs) [96], as their application can significantly reduce RNN model parameters by up to 1000 times. However, they face precision–tradeoff issues in CNN compression. Nevertheless, the authors of [97] proposed a TT format suitable for CNN models. Additionally, TT decomposition was found to be practical for compressing three-dimensional CNNs (3DCNNs) in [98], where an innovative approach for selecting TT ranks to maximize compression gains was outlined. Another pioneering approach involves utilizing TR decomposition, as shown in [99], which employs a progressive genetic algorithm for optimizing rank selection.

In summary, CP decomposition represents tensors as the sum of first-order tensors, offering a unique perspective on compression. On the other hand, Tucker decomposition breaks down tensors into groups of matrices and a compact core tensor. TT decomposition constructs complicated three-dimensional tensor structures, particularly adept at handling high-order tensors. TR decomposition is an extension of TT decomposition, emerging as a linear combination of TT decompositions. These tensor decomposition techniques continue to advance the efficiency of EEG signal processing.

### 3.4. Knowledge Distillation

Knowledge distillation is a technique that transfers knowledge from large, complex CNNs (teachers) to smaller, more efficient CNNs (students). The student network is trained to mimic the behavior of the teacher, resulting in a compact model with comparable performance. This concept can be traced back to the pioneering work in [101] and has since been extended to the context of deep learning [102]. The core challenge of knowledge distillation revolves around the techniques used to transfer knowledge from the teacher model to the student model, which involves three fundamental components—knowledge, distillation algorithms—and the architecture defining the relationship between the teacher and student models. In this context, knowledge manifests in various forms, including logits, activations, or features extracted from intermediate layers of the teacher model. The distillation algorithms can be categorized as offline, online, or self-distillation.

Offline distillation [101,103,104,105] extracts knowledge from a pre-trained teacher model and uses the soft label output of the teacher model to train the student network. The authors of [103] used data augmentation to exploit the output distributions of multiple teacher networks. The authors of [101] introduced a tailored distillation approach for quantized models, demonstrating that quantized student networks can closely match the accuracy of full-precision teacher networks while achieving high compression rates and inference acceleration. In contrast, [104] pioneered a data-free technique, training the student network using synthetic data responses from the complex teacher network. Online distillation [106,107,108] occurs during the simultaneous training of both the teacher and student models. It employs online knowledge distillation using the soft-label outputs of the teacher network. Ref. [106] proposed training the student model at different checkpoints of the teacher model until convergence is achieved. Meanwhile, Collaborative Learning Knowledge Distillation (KDCL) [107] dynamically generates high-quality soft targets through an ensemble approach for one-stage online training.

Furthermore, as observed in [105,108], knowledge distillation extends its influence to the intermediate layers of the teacher network. Layer Selective Learning (LSL) [105] provides a two-layer selection scheme, known as the inter-layer Gram matrix and layered inter-class Gram matrix, enabling the selection of intermediate layers in both the student and teacher networks for knowledge distillation. In [108], an updated ensemble-based knowledge distillation method was used to leverage features from intermediate layers, facilitating the simultaneous training of the distilled student system and the ensemble of teachers without the need for pre-trained teacher models.

On the other hand, an efficient neural network architecture design for self-knowledge distillation without training a large teacher network has been developed [109,110]. The authors of [109] introduced an auxiliary self-learning network that transfers refined knowledge to the classifier network using soft labels and intermediate feature maps. In [110], effective augmentation strategies were employed to extract knowledge using the best-performing student network from past epochs while simultaneously training the current student network, thereby enhancing network generalization.

In knowledge distillation, hardware constraints can be incorporated during the teacher model’s training of the student model to guide it toward meeting specific hardware performance goals. The authors of [111] used knowledge distillation to develop NEMOKD. This system balances latency, accuracy, training time, and hardware costs. It was evaluated on the Movidius Myriad X VPU and Xilinx’s programmable Z7020 FPGA. This hardware-aware distillation process minimizes memory usage and computational load, making it possible for the student model to run efficiently on wearable EEG devices.

### 3.5. Neural Architecture Search

Neural Architecture Search (NAS) is an automated approach for discovering the optimal neural network architecture. NAS defines a search space that includes various CNN models and performance evaluation algorithms. It generally does not require human intervention to discover the most efficient CNN model for specific needs. Unlike methods such as pruning, quantization, and tensor decomposition, as discussed above, NAS requires a specific model. Early NAS methods [112] initially generated a multitude of candidate network architectures based on predefined search space criteria. Each architecture undergoes rigorous training until convergence and is then ranked based on its accuracy on a validation dataset. These rankings provide feedback for fine-tuning the search strategy and generating new neural architectures. This iterative process continues until specified termination conditions are met. Ultimately, the most promising architectures are meticulously selected and evaluated using a test dataset. Since the search space contains thousands of neural networks, training and evaluating each network incurs significant computational and time costs.

Researchers need to find ways to identify efficient network architectures and reduce the computational costs required for training and evaluation. Current NAS methods typically involve three primary stages: training a super network, training and evaluating sampled networks, and finally, improving the discovered networks.

#### 3.5.1. Search Space

The complexity increases exponentially when searching for networks and their interconnected structures in the NAS space [112]. Therefore, the search space needs to be limited. A limited number of network structures are used to form iterative and repeated units. Examples of this approach include NASNet [113], ENAS [114], and AutoDispNet [115]. For instance, AutoDispNet focuses on finding two different cell types, namely normal cells and reduction cells, within a predefined overall network framework. In contrast, ENAS involves an approach where all sub-models share weights collectively, significantly increasing GPU processing time but achieving over a thousandfold improvement compared to traditional NAS [112].

Depending on the specific task, we hope to find a balance between latency (inference time) and memory usage while maintaining satisfactory accuracy. MnasNet [116] is a platform-aware NAS tailored for mobile devices. It studies complex network architectures and analyzes the optimal trade-off between latency and accuracy.

#### 3.5.2. Search Algorithm

In the complex field of NAS, search strategies shape the exploration of a specified search space. Typically, NAS involves a generator that generates sample architectures and an evaluator responsible for assessing their post-training performance. Various search algorithms, such as reinforcement learning (RL) and neural evolution algorithms, have emerged to address this essential issue. RL has achieved significant success, with NAS-RL [112] and MetaQnn [117] demonstrating classification accuracy in image classification tasks. Neuroevolution algorithms originally covered the optimization of neural structures and weights [118], but recently, the focus has shifted primarily toward optimizing neural structures [119]. Ref. [119] showed that both reinforcement learning and neural evolution consistently outperform random search (RS) in terms of final test accuracy. Specifically, neuroevolution consistently manages to strike a balance between high performance and compactness.

#### 3.5.3. Performance Evaluation Strategy

The strategy for evaluating neural architecture performance is a fundamental aspect of the NAS field. In the early days of NAS, the approach was to assess the quality of each sampled network through exhaustive training from scratch [112]. However, this process proved to be highly costly in terms of computational resources and time. Recent advancements have introduced innovative techniques to streamline the performance evaluation step while maintaining high-quality results. Progressive NAS [120] and ReNAS [121] have ingeniously addressed this challenge by constructing accuracy predictors. These predictors are trained using data collected from a subset of sampled network candidates within the search space. During the search process, the trained predictor can evaluate the searched network at no additional cost. Another approach is known as one-shot NAS [122]. It reduces the evaluation cost by training a super network. Each sampled network inherits weights from the super network so that no additional training cost is required. This weight-sharing mechanism is at the core of one-shot NAS and plays a crucial role in reducing the computational burden associated with NAS, thereby significantly improving efficiency.

Automatically designed networks are very powerful but are not suitable for deployment on wearable EEG devices. Recognizing this challenge, hardware-aware NAS methods [123,124] have recently emerged. Unlike previous search strategies, they incorporate hardware factors, such as inference time, power consumption, and memory consumption, into the NAS search process. Ref. [125] proposed a two-stage automated hardware-aware search algorithm, which was evaluated under identical latency constraints across CPU, DPU, and VPU platforms. Compared to manually designed search spaces, this approach achieved higher accuracy. Ref. [126] introduced MemNAS, a search method that takes memory requirements into account. By progressively growing and pruning the network, this algorithm maintains a balance between accuracy and memory consumption across devices with varying resource constraints. To further reduce evaluation costs, hardware-aware NAS focuses on training a single network that can adapt to different hardware architectures, achieving a balance between high performance and cost-effectiveness.

#### 3.5.4. EEG-Based Convolutional Neural Network Hardware Acceleration

In this subsection, we discuss the available hardware architectures suitable for CNNs. Next, we briefly review some cases showing representative CNN accelerators for EEG signal processing. Over the past decade, CNN hardware acceleration technology has developed significantly, mainly due to the emergence of high-performance GPUs, which are able to cope with the increasing memory requirements and computational complexity of CNNs. Although early applications of deep learning mostly focused on large computing platforms, there is now an increasing need to deploy CNNs on edge devices with limited hardware resources and energy, such as portable or wearable EEG signal-processing devices. To this end, hardware solutions for CNNs have expanded from general-purpose architectures (such as CPUs and GPUs [127]) to spatial architectures (such as FPGAs [22,128] and ASICs [129,130]).

The authors of [128] described the implementation of the VGG model on a PYNQ Z1 FPGA development board for motor imagery tasks. Figure 1 shows a CNN design based on a Xilinx FPGA. The authors found that quantizing from a 32-bit floating-point format to a 16-bit fixed-point format reduced BRAM and DSP resource usage by 25%, with no significant loss in accuracy. In contrast, quantizing to an 8-bit fixed-point format resulted in a 7% decrease in accuracy. The authors of [130] implemented the STFT and CNN models on a TSMC 28nm chip and tested them using the DEAP dataset for emotion recognition during movie-watching tasks.

The authors of [129] used a 180 nm 1P6M CMOS as a coprocessor to accelerate the CNN inference. The accelerator achieved 97.8% accuracy in floating-point operations and 93.5% accuracy in fixed-point operations. As shown in Figure 2, the CNN is transferred to the edge device after training. The authors also defined a coprocessor interface (CCI) to facilitate communication between the RISC-V core and the accelerator.

In the context of portable EEG signal processing, FPGAs have become a mainstream choice for research. The reason is that the architectural design of FPGAs is very suitable for this purpose, and their unique flexibility and efficiency provide high computing performance with low power consumption. In [127], the authors compared three platforms—an Intel Core i7-7500U CPU, an NVIDIA Jetson Nano GPU, and a Xilinx Ultrascale+ ZU7EV FPGA—for motor imagination tasks. As shown in Figure 3, in terms of FOM and energy consumption rankings, the FPGA may be the best solution, followed by the Jetson Nano.

In [9], the authors proposed a software and hardware co-design solution for a lightweight CNN. As shown in Figure 4, they proposed another FPGA acceleration implementation. Unlike [128], the calculation results of each layer are transmitted back to the ARM processor through the AXI bus. This implementation method can flexibly configure various parameters of the CNN to adapt to rapidly evolving CNN technology but at the cost of increased data pressure. The results of the eye-state recognition task were evaluated on three platforms. As shown in Figure 5, FPGA has obvious advantages in weight, cost, and power consumption. In contrast, it is slightly less efficient than a high-performance PC in terms of energy consumption.

The authors of [22] introduced BioCNN, which was designed for emotion classification tasks. BioCNN was implemented on a Digilent Atlys board with a low-cost Spartan-6 FPGA and tested using the FEXD and DEAP datasets. It achieved an energy efficiency of 11 GOPS/W and a latency of less than 1 ms, significantly below the standard 150 ms threshold for human–computer interaction and approaching the performance of software-based solutions. Compared with the GPU, the FPGA not only accelerates the CNN inference process by processing a large number of multiply-accumulate (MAC) operations in parallel but also optimizes memory access and data flow processing through customized hardware logic design. Because the FPGA’s processing elements (PEs) have local storage and control logic, they can communicate directly with each other with minimal memory bandwidth requirements, significantly increasing data reuse. This feature is particularly critical for the low-power and efficient computing requirements of portable EEG devices.

In contrast, although GPUs perform well in the field of general computing, in the literature we surveyed, most of them were implemented in high-power PCs. Additionally, their low customization capabilities limit their application on portable devices. ASICs, on the other hand, although capable of delivering extreme performance and efficiency for specific applications, are generally only suitable for specific applications in mass production due to their high development costs and lack of flexibility. Therefore, FPGAs have become the preferred hardware platform for CNN acceleration in portable EEG signal-processing devices due to their low power consumption, flexibility, and sufficient computing power.

Table 3 summarizes the state-of-the-art CNN hardware solutions discussed in this section.

## 4. Challenges and Opportunities

### 4.1. Standardized Evaluation Paradigms

Due to the different EEG application tasks, different indicators are needed to evaluate their performance. These indicators are diverse and complex. Due to the lack of standardized performance evaluation paradigms, not all EEG signal-processing hardware solutions provide the same indicators. Examples of these indicators include the following:**Wearability:** Wearability is an important feature in tasks that require constant, unobtrusive, real-time monitoring, such as sleep tracking or emotion classification. Uncomfortable or bulky devices may discourage users from consistently wearing them, potentially affecting data quality and continuity.**Energy Efficiency and Area**: Different hardware architectures cannot usually be directly compared. For example, FPGAs typically consume more power than ASICs. Moreover, the power consumption of an FPGA largely depends on its evaluation board. In other words, even the same FPGA design may lack comparability. For example, Ref. [131] introduced a normalized energy metric for comparing the power consumption of different technologies. The formula is as follows:
(6)$${Normalized}_{Energy}=\frac{Power\;\times \;ExecTime}{N_{Class}{\left(\frac{V_{DD}}{V_{ref}}\right)}^{2}}\;$$
where the unit of normalized energy is nJ and $N_{Class}$ is the number of classes.**Latency**: Latency is the time it takes for the system to produce classification results after receiving an EEG epoch. For instance, iMotor imagery tasks are highly dynamic, so shorter delays are desirable in such tasks. Subjects are less likely to maintain high levels of attention and exhibit consistent responses over longer time intervals. This contrasts sharply with seizure detection devices based on EEG, where longer detection time windows are more effective, as relevant seizures become sparser over time [132].

Standardized metrics and evaluation paradigms help to fairly compare different hardware solutions, promote healthy competition, and accelerate technological progress. For techniques such as compression, quantization, and hardware acceleration, standardized metrics help to explore the best efficiency for specific applications.

### 4.2. High-Quality EEG Datasets

Acquiring high-quality EEG datasets is a formidable challenge, significantly impacting the robustness assessment of systems. For DL networks, such as CNNs, that require large and diverse data, the quantity and quality of EEG data affect their training results. Unfortunately, most available EEG datasets lack the approval needed as benchmark resources to validate EEG-based classifiers in medical settings. Therefore, standardization of data collection or involving more subjects in the dataset development process can promote the development of EEG signal-processing technology, both in software and hardware.

Therefore, datasets must provide raw EEG data along with comprehensive data acquisition details. For instance, [133,134] used bandpass Hamming windows as filters. In contrast, [135] employed built-in Sinc filters to eliminate frequency components above specified thresholds, and [131] used a high-pass filter to remove unwanted event codes and noise channels from the raw input data. Equally important is defining stimulus types before EEG data collection, with common choices being auditory, visual, and audiovisual mixed stimuli. However, in a particular task, the choice of stimuli is determined subjectively by the researcher, making it difficult to replicate the classification results. Stimulus selection may become one of the challenges of EEG signal acquisition.

Reporting other subject-related information, such as race, occupation, personality traits, and physiological conditions, as well as comprehensive experimental setup details, is crucial for identifying external influences. Reference sensors in EEG data collection are also vital for anchoring data points. Some datasets [136] use single-modal approaches (only EEG data), but multimodal datasets [133,137], including ECG, facial video clips, peripheral physiological signals, and infrared spectra, offer a more comprehensive perspective and additional classification strategies.

In particular, CNNs can seamlessly integrate these multi-modal data sources to enhance the model’s generalization ability. In conclusion, future EEG-based datasets for standardized benchmarks should encompass multi-modality, universality, consistent labeling, and preprocessing libraries, among other features.

### 4.3. Embedded Preprocessing

As previously mentioned, the removal of artifacts can directly impact the performance of CNNs. These artifacts often mask meaningful neural activity. For instance, power-line noise can obscure cortical activity around 50 Hz (or 60 Hz), while electrode drift can disrupt slow cortical potentials. When eye- or muscle-related activity occurs simultaneously with external stimulation, these artifacts may be mistaken for cortical activity [138]. This is a common situation in EEG data collection experiments. However, artifact removal can sometimes introduce unintended side effects, such as the multiplicative offset triggered by high-pass filters, which may occasionally be misinterpreted as neural activity [139]. Filter-based preprocessing alone cannot eliminate all artifacts from EEG data. The design of EEG-based hardware systems also needs to be combined with other efficient preprocessing techniques. While the ICA-based techniques mentioned earlier are effective, they are not entirely hardware-friendly and may not comprehensively address all types of artifacts. For instance, ICA cannot distinguish artifacts originating from electrode-specific or sensor-specific sources [140]. Therefore, finding and developing solutions that can effectively handle various artifacts while maximizing the use of hardware characteristics remains a key direction for achieving real-time classification of EEG signals.

### 4.4. High-Resolution Feature Extraction

The concept of EEG frequency bands, although valuable in engineering applications for enhancing classification accuracy, lacks comprehensive neurophysiological underpinning. Relying on PSD as a proxy for extracting EEG signals representing brain activity is an effective but somewhat rudimentary approach. Integrating CFC features into EEG-based hardware classifiers presents a promising alternative for wearable devices. This approach has the potential to enhance robustness and reduce memory requirements by emphasizing connectivity graphs rather than storing complete raw EEG datasets [141].

In the field of frequency-domain features, another avenue for improvement involves systematically incorporating spatial asymmetries. As discussed in Section 2.2.4, extending this practice to other EEG electrode configurations can further refine our understanding of EEG wave patterns and their impact on CNN-based EEG classifiers in both software and hardware implementations.

## 5. Conclusions

This review focuses on preprocessing, feature extraction hardware solutions, and acceleration techniques for CNNs analyzing EEG signals. It encompasses a wide range of applications, including emotion classification, motor imagery, seizure detection, and sleep monitoring. This review begins with a detailed analysis of hardware implementation techniques for EEG signal preprocessing and five types of feature extraction: frequency-based, time-based, time-frequency, spatial features, and raw signals. Subsequently, efficient CNN solutions based on EEG signals are elaborated, including pruning, quantization, tensor decomposition, knowledge distillation, and NAS. In the following sections, this review addresses the challenges and opportunities associated with EEG hardware accelerators, such as multi-modal universal datasets, standardized metrics, embedded preprocessing, and high-resolution feature extraction. Therefore, this article summarizes existing research results in the field of EEG signal-processing hardware implementation by reviewing the current literature. It also identifies unresolved problems and research gaps and provides future research directions for other researchers.

## Figures and Tables

**Figure 1 sensors-24-05813-f001:**
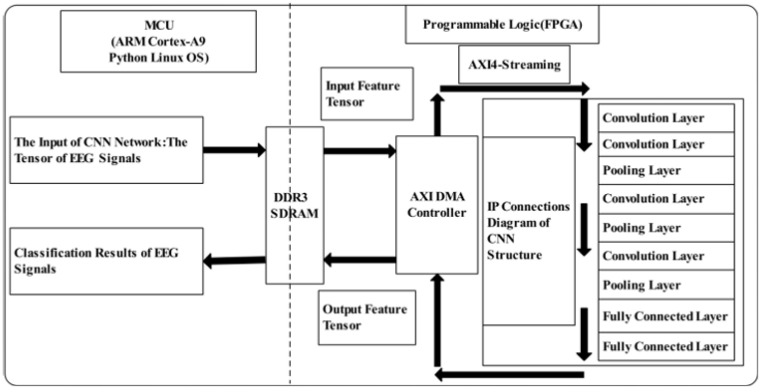
FPGA block diagram from [128].

**Figure 2 sensors-24-05813-f002:**
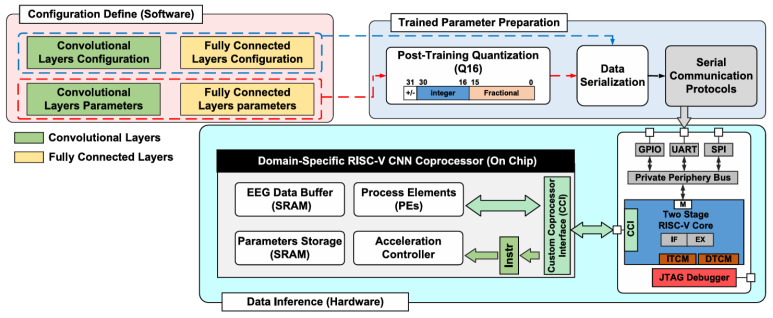
ASIC block diagram from [129].

**Figure 3 sensors-24-05813-f003:**
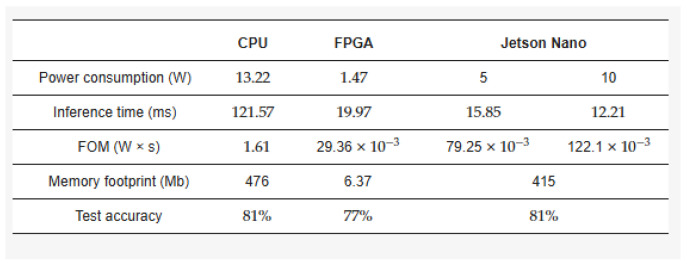
Results on different platforms from [127].

**Figure 4 sensors-24-05813-f004:**
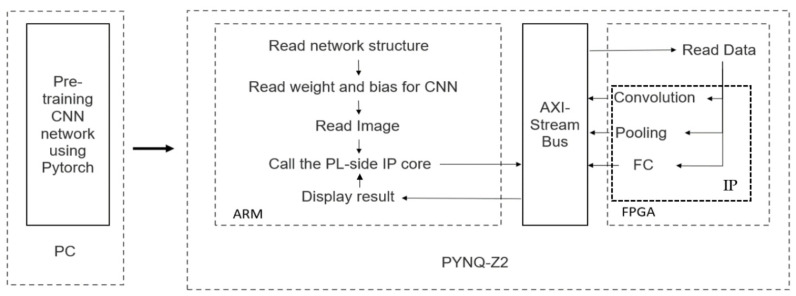
FPGA block diagram from [9].

**Figure 5 sensors-24-05813-f005:**
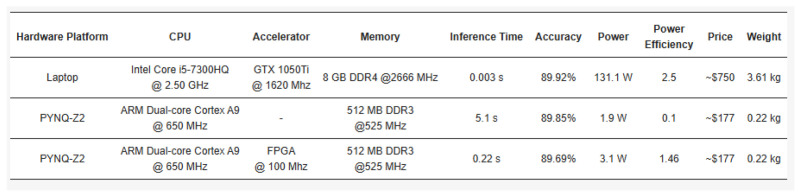
Results on different platforms from [9].

**Table 1 sensors-24-05813-t001:** Precision and recall scores for the proposed model.

EEG Band	Frequency (Hz)
Alpha	8–13
Beta	12–30
Delta	0.5–3.5
Theta	4–7
Mu	8–12
Gamma	26–100
**Artifacts**	**Frequency (Hz)**
ECG	0.5–40
EOG	0.1–17
EMG	0.1–2.5/20–100
RS	0.1–2.5
Ep	0.1–35
Cable	2–15
AC	50

**Table 2 sensors-24-05813-t002:** Feature extraction methods.

Feature Domain	Authors	Algorithm	Work Description
Frequency-based	[18,21]	PSD	A power spectrum approximation per channelHardware friendly, no FFT accelerator requiredEasily loses band information
	[22]	PSD	Dataset: Own FEXDHardware: Digilent Atlys with Spartan-6An FFT accelerator with 128 frequency binsCaptures fluctuations within and between frequency bands
	[31]	CFC	Uses connection diagrams to represent EEGReduces the need to store and compute EEGCORDIC required Hilbert transform of each frequency band
Time-based	[19]	HOC, SK	Hardware: TSMC 0.18 um 1P6M CMOS chipDatabases: DEAP and SEEDUses ASKI to reduce gate counts, 3-bit SR triggers
	[32]	Hjorth	Hardware: Zynq-7000 FPGADataset: Bonn University EEG databaseClassifier: KNN
	[33,34]	SE	For statistics, complexity, and irregularity systemsFor analyzing short and noisy datasetsRelatively hardware-friendly
Time-frequency	[36]	STFT	Hardware: Xilinx ZedboardExtracts 20 frequency-band features from 128 channels
	[37]	STFT	Used for epilepsy detectionOnly simulation; not applied to real FPGA chips
	[38]	STFT	Hardware: ADVANTEST V93000 PS1600 chipDataset: DEAPSTFT features mixed with asymmetric indices
	[39]	DWT	Hardware: Basys 3 Artix-7 FPGA BoardClassifier: LDASuitable for developing low-cost, marketable products
	[40]	CWT	Hardware: Spartan 3AN FPGAEfficient architecture in Fourier space via optimized speed and silicon area
Spatial	[41]	CSP	Hardware: Altera FPGA platform (Stratix-IV)Dataset: motor imagery tasks in a BCI competitionClassifier: MDTime delay of 0.5 s
	[42]	CSP	Improvement on method in [41]; reduces the delay to 0.399 s
	[43]	SCSSP	Hardware: Virtex-6 FPGADataset: BCI Competition IV-dataset 2aClassifier: SVM
	[18]	Ai, PSD	Based on a scaled version of IHPR with customized LUT
	[21]	Ai, PSD	Improvement on method in [18]; reduces the gate and power consumption
	[45]	DTF	Describes incidental effect between two channelsDifficult to implement in hardware
Raw	[50]	PSDCNN	Dataset: Own For emotion recognition and fatigue-driving

“Ai” denotes asymmetric indices.

**Table 3 sensors-24-05813-t003:** List of reviewed CNN hardware techniques.

Method	Authors	Work Description
Weight Pruning	[51,52]	Hessian matrix of the loss function
	[53,54]	Three-step method
	[57]	Frequency-domain approach
	[51]	Variable compression rates
	[58]	GMM
	[59]	Energy reduction with accuracy loss
Structural Pruning	[60,61,62]	Criterion-based pruning
	[63]	Ranking-based pruning
	[64,65]	Optimization-based pruning
	[66,67,68]	Whole-layer pruning methods
	[69,70,71]	Automatically identifies and prunes
Quantization	[72]	Characterized by even step sizes
	[73]	Represented by logarithmic distributions
	[75,76]	Post-training quantization
	[77,78,79]	Quantization-aware training
	[80,81,82]	Binary quantization networks
	[72,82,83,84,85]	Optimized binarization techniques
LRM Decomposition	[88]	Decomposes the product of weight matrices and input data
	[89]	Sparsity: Maintaining a lower rank for less critical neurons
	[90]	Channel-wise SVD decomposition by dividing kernels into two layers
	[91]	Low-rank approximation without invoking SVD
	[92]	Joint matrix factorization scheme
Tensorized Decomposition	[93]	Tucker decomposition
	[94,95]	CANDECOMP/PARAFAC (CP)
	[96,97,98]	Tensor train decomposition
	[99]	Tensor ring decomposition
Knowledge Distillation	[101,103,104,105]	Offline distillation
	[106,107,108]	Online distillation
	[109,110]	Self-knowledge distillation
NA Search Space	[113]	NASNet
	[114]	ENAS
	[115]	AutoDispNet
	[116]	MnasNet
NA Search Algorithm	[112]	NAS-RL
	[117]	MetaQnn-RL
	[119]	Optimizing neural structures
NA Search Performance	[120]	Progressive NAS
	[121]	ReNAS
	[122]	One-shot NAS
	[123,124]	Hardware-aware NAS

“NA” means neural architecture. “LRM” means low-rank matrix.

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
