# Peer review of "A Comprehensive Review of Hardware Acceleration Techniques and Convolutional Neural Networks for EEG Signals"

_sensors, 2024, doi:10.3390/s24175813_

Round 1

Reviewer 1 Report

Comments and Suggestions for Authors

The paper demonstrated an extensive review of EEG feature-extracting methods and convolutional neural network (CNN) implementations for EEG signals. Despite covering a lot of techniques, I find the body of the paper confusing and not correlating with the title and the abstract. The paper is not suitable for publication and I highly recommend the author reconstruct the paper to address the following comments:

  1. In Table 1, “Alfa” band is not the correct name, rename it to “Alpha”. Mu and alpha bands are very similar; I recommend the author mention which feature extraction method can distinguish these two bands.

  2. The paper should specify what kind of hardware they are dealing with and why hardware accelerators are important to them. Is it an embedded system or a wearable device? If the method can be deployed to a cloud server with reasonable resources, hardware acceleration will not be a big concern. 

  3. The author seems to diverge from the main topic of the paper when discussing methods for reducing the complexity of the deep learning models and feature extractions relying heavily on statistics or linear algebra and does not have any hardware-specific implementation, such as ICA, EVD or CFC. If the authors really want to mention these methods, they should focus on the implementation of these methods and explain how these methods can be tuned to accelerate the implementation on specific hardware.

  4. The authors should list what hardware accelerators are available such as mix-precision on NVIDA GPUs, TPU or XLA and discuss how these can be incorporated into EEG signal processing.

  5. The topic of the paper is hardware acceleration and CNN. The flow of the paper is confusing to me. The hardware acceleration for CNN was briefly discussed and the authors moved on to neural architecture search and knowledge distillation which do not have any relation to hardware acceleration. The author mentioned a lot of methods around CNN but did not discuss how these could be used for EEG signals.

  6. High quality dataset is important but how is it related to hardware acceleration?

Comments on the Quality of English Language

The paper demonstrated an extensive review of EEG feature-extracting methods and convolutional neural network (CNN) implementations for EEG signals. Despite covering a lot of techniques, I find the body of the paper confusing and not correlating with the title and the abstract. The paper is not suitable for publication and I highly recommend the author reconstruct the paper to address the following comments:

  1. In Table 1, “Alfa” band is not the correct name, rename it to “Alpha”. Mu and alpha bands are very similar; I recommend the author mention which feature extraction method can distinguish these two bands.

  2. The paper should specify what kind of hardware they are dealing with and why hardware accelerators are important to them. Is it an embedded system or a wearable device? If the method can be deployed to a cloud server with reasonable resources, hardware acceleration will not be a big concern. 

  3. The author seems to diverge from the main topic of the paper when discussing methods for reducing the complexity of the deep learning models and feature extractions relying heavily on statistics or linear algebra and does not have any hardware-specific implementation, such as ICA, EVD or CFC. If the authors really want to mention these methods, they should focus on the implementation of these methods and explain how these methods can be tuned to accelerate the implementation on specific hardware.

  4. The authors should list what hardware accelerators are available such as mix-precision on NVIDA GPUs, TPU or XLA and discuss how these can be incorporated into EEG signal processing.

  5. The topic of the paper is hardware acceleration and CNN. The flow of the paper is confusing to me. The hardware acceleration for CNN was briefly discussed and the authors moved on to neural architecture search and knowledge distillation which do not have any relation to hardware acceleration. The author mentioned a lot of methods around CNN but did not discuss how these could be used for EEG signals.

  6. High quality dataset is important but how is it related to hardware acceleration?

Reviewer 2 Report

Comments and Suggestions for Authors

The author did a good job. However, the suggestions that follow are for the author to consider in order to improve the manuscript's quality.

1. The abstract mentions "diverse applications" of EEG signals, but provides only a few concrete examples. Please provide a more thorough list or generalize the applications to better fit the phrase "diverse"?

2. The abstract ought to clearly convey the review's unique contributions and distinctiveness in comparison to existing literature. What specific shortcomings does this review address?

3. It would be beneficial to give a comparative study of the hardware options described. How do these solutions compare to one another in terms of efficiency, cost, and applicability?

4. While the research explores hardware acceleration solutions, the difficulties of applying these techniques in real-world applications are not thoroughly investigated. What are the practical barriers, and how can they be overcome?

5. In the context of CNN architectures, numerous strategies are mentioned, including pruning and quantization. Please include case studies or examples of how these techniques have been used especially to EEG data.

Comments on the Quality of English Language

Please revise the manuscript for language editing and errors.

Round 2

Reviewer 1 Report

Comments and Suggestions for Authors

The authors have replied to all of my comments. I still have a few concerns:

  1. There is no comparison between CNN and the non deep learning approach. The author suddenly jumped to CNN without any discussion on why CNN is a better choice. The non deep learning approaches are simple and fast and can be easily implemented in embedded systems. The accuracy of these methods might not be good due to the complexity of EEG where the signal is continuous (not discrete). CNN is a deep learning approach where the number of parameters are huge and hardware acceleration is very useful in this case. More discussions around this are needed in the paper.

  2. In the rebuttal, the author said NAS and knowledge distillation could be ‘hardware-aware’. Yet, this is not discussed in the paper. The author should discuss how these methods can be ‘hardware-aware’.

  3. Reference format is not correct throughout the paper. Please check the guidelines from MDPI.

Reviewer 2 Report

Comments and Suggestions for Authors

Accepted

Author Response

We sincerely thank you for your feedback.